# Plant-Derived Smoke Affects Biochemical Mechanism on Plant Growth and Seed Germination

**DOI:** 10.3390/ijms21207760

**Published:** 2020-10-20

**Authors:** Amana Khatoon, Shafiq Ur Rehman, Muhammad Mudasar Aslam, Muhammad Jamil, Setsuko Komatsu

**Affiliations:** 1Department of Botanical & Environmental Sciences, Kohat University of Science & Technology, Kohat 26000, Pakistan; proteomics.sp@gmail.com; 2Department of Biology, University of Haripur, Haripur 22620, Pakistan; drshafiq@yahoo.com; 3Department of Botany, University of Science & Technology Bannu, Bannu 28100, Pakistan; mudasar_kust@yahoo.com; 4Department of Biotechnology & Genetic Engineering, Kohat University of Science & Technology, Kohat 26000, Pakistan; jamilkhattak@yahoo.com; 5Department of Environmental and Food Sciences, Fukui University of Technology, Fukui 910-8505, Japan

**Keywords:** plant-derived smoke, biostimulant, omics, seed germination, plant growth

## Abstract

The role of plant-derived smoke, which is changed in mineral-nutrient status, in enhancing germination and post-germination was effectively established. The majority of plant species positively respond to plant-derived smoke in the enhancement of seed germination and plant growth. The stimulatory effect of plant-derived smoke on normally growing and stressed plants may help to reduce economic and human resources, which validates its candidature as a biostimulant. Plant-derived smoke potentially facilitates the early harvest and increases crop productivity. Karrikins and cyanohydrin are the active compound in plant-derived smoke. In this review, data from the latest research explaining the effect of plant-derived smoke on morphological, physiological, biochemical, and molecular responses of plants are presented. The pathway for reception and interaction of compounds of plant-derived smoke at the cellular and molecular level of plant is described and discussed.

## 1. Introduction

Plant-derived smoke is a well-known agent for promoting plant growth and development [1]; and positively affects plant species from various habitats [2,3]. The seed germination cues associated with fire or post-fire environments were identified as heat, temperature, chemicals, and smoke [4]. On the other hand, smoke produced during fire was recognized as the major germination cue in post-fire environments [1,2]. Despite its potential as a post-fire cue for seed germination, burning of vegetation has many disadvantages. The negative impacts of the fire and smoke include killing the beneficial soil insects and microorganisms, loss of various minerals, and air pollution [5]. The major factor contributing to air pollution due to vegetation burning is the addition of CO_2_ to the environment. Species related to a fire-prone environment positively respond to smoke produced during fire in the forest [6]. Plant-derived smoke enhanced the germination process in various plant communities such as the South African Mediterranean [7,8] and Californian chaparral [9,10]. Compounds in smoke are stable at high temperatures, water soluble, and very active even at low concentrations [11]. The long-lasting effectiveness of plant-derived smoke solution ruled out the uncertainty about its storage period and made it more worthy in terms of its in-time availability.

Plant-derived smoke was proven to be a promoting factor for several growth-related phenomena of plants including breaking seed dormancy, accelerating seed germination, and increasing seedling vigor [12]. Biochemical parameters such as photosynthetic pigments, total nitrogen, total soluble proteins, and photosynthetic rates increased under plant-derived smoke treatments [13]. Furthermore, a supply of plant-derived smoke solutions during flooding stress led to the flooding recovery of soybean after the removal of water [14]. It reduced the inhibitory effects of heavy metal, drought, salinity, and high/low temperature stresses on plant growth [15]. These results indicated that plant-derived smoke is a growth enhancer, reducer of inhibitory effect of environmental stresses, and a possible plant-growth fertilizer.

After the authentication of plant-derived smoke as an important germination cue and a positive growth regulator in various plant growth-related wonders [1,2], various arguments were made about the nature of smoke solution, different active compounds present in smoke, and the nature/mode of action of smoke. This review highlights the roles of plant-derived smoke in seed germination and plant growth, summarizes its role in physiological parameters of plants, and focuses on molecular aspects of plant responses towards smoke.

## 2. Preparation of Plant-Derived Smoke

Plant-derived smoke is prepared through various methods, which are convenient and economical to adopt. The most familiar way to produce plant-derived smoke solution is to bubble out the smoke through water to dissolve the biologically active compounds of smoke in water. Smoke was generated in a drum and bubbled through distilled water using compressed air. After dilution, this smoke extract was applied to seeds of various plant species resulting in improved seed germination. Based on this method, a wide range of plant materials were used to prepare aqueous smoke extracts [16]. All plant materials are generally suitable for preparation of smoke extracts [17]. A concentrated aqueous smoke extract can be diluted with water.

The active compounds in smoke are not carried far in smoke. These compounds condense as the smoke cools and remain close to the fire site [18]. The method was further calculated using a standard weight of the semi-dried plant material weighing 333 g burnt in chimney, using an electric heater. Smoke was produced and passed through a beaker with 1 L distilled water in order to obtain a concentrated form of smoke solution. The concentrated plant-derived smoke was filtered (Figure 1). This standard concentrated smoke solution was used in various dilutions in different studies. The variable effect of plant-derived smoke solution in different dilutions is an interesting factor. It is assumed, that due to presence of numerous compounds in plant-derived smoke solution, the concentration balance among these compounds is important and is specific for each plant species. This might be the reason that different plants show optimum growth response at different dilutions which support our assumption.

## 3. Components in Plant-Derived Smoke

Plant-derived smoke is a complex mixture of compounds derived from burning a natural mixture of plant species and stimulates seed germination and plant growth from a wide range of plants. As compounds in plant-derived smoke, butanolide known as karrikins and cyanohydrin are the active compound (Table 1 and Figure 2). The active compounds in plant-derived were separated from different plants, showing the diverse nature of smoke based on the plant used to produce smoke. These compounds set the base that why different plants respond to plant-derived smoke differently.

### 3.1. Karrikins

After the discovery of plant-derived smoke as a vital germination agent [1], the active compound from burning filter paper and cellulose-derived smoke was identified using a bioassay-guided fractionation procedure which includes solvent partitioning, high-performance liquid chromatography, gas chromatography–mass spectrometry, and nuclear magnetic resonance [11,25]. This compound stimulated germination of different smoke-responsive species from various regions including Australia, North America, and South Africa [26]. The structure of the promoting compound was confirmed by chemical synthesis as 3-methyl-2*H*-furo[2,3-c] pyran-2-one (Figure 2) [11] which was a class of compounds containing a butenolide (a four-carbon lactone) fused to a pyran ring. This compound was named as karrikinolide (KAR1). It was active at very low concentrations even down to 10^−10^ M [11]. Structurally, karrikins (butenolide) consist of a five-membered butenolide ring fused to a six-membered pyran ring (Figure 2). They are different with respect to methyl substitutions, in which KAR1 and KAR3 are the most active for stimulating seed germination [27]. Various analogs of karrikins were synthesized by substitutions at carbons 3, 4, and 7 [25,28,29,30]. The activities of these analogs were also evaluated on highly smoke-responsive species including lettuce, which clarified that plant responses to various analogs varies [27]. These analyses were performed at seed germination level; however, extending these investigations to post-germination stages would further clarify which analog is most suitable for the optimum response of a specific bioactivity.

Karrikins trigger the germination of numerous species from fire-prone [31], fire free [32], and agricultural environments [3,25]. Besides having promoting effects on seed germination, karrikinolides are phytoreactive compounds with applications in horticulture, ecological restoration, and agriculture [31]. Plant-derived smoke significantly enhanced the wheat seedling growth with a remarkable increase in leaf area, lengths, and fresh weights of root and shoot in wheat [33]. Smoke-water-treated tomato seedlings exhibited a remarkable increase in growth and yield [34]. Seedling mass was reported to be enhanced by smoke treatment in lettuce and cucumber [17,35]. Plant-derived smoke showed a promoting effect on edible banana by increasing the number of shoots [36], while, in onion, number, length and mass of leaves, and bulb diameter and weight was also increased in response to smoke treatment [35]. Growth enhancement in horticultural crops suggested the possible use of smoke technology for cultivation and improvement of these crops. Cytotoxic and genotoxic effects of smoke-water were evaluated with five strains of *Salmonella typhimurium* marking it safe at the concentrations tested between 1 × 10^−4^ and 3 × 10^−10^ M [37]. Moreover, in addition to promote the early growth, KAR1 enabled the tomato plants to tolerate temperature stress [38]. A significant improvement in the adverse effects of salt stress was observed in smoke-treated maize seedlings [39]. Salt tolerance was attributed to increased activities of antioxidants, increased level of K^+^ and Ca^+2^, while the Na^+^ content was reduced in smoke-treated maize under salt stress [39]. A similar trend of stress alleviation was observed in smoke primed rice seedlings under salt stress. In addition to previously discussed parameters, alleviation was further supported by increase in cell membrane stability of smoke primed rice seedlings compared to non-treated ones [40]. Furthermore, flooding stress tolerance [14] and reduction in the inhibitory effects of heavy metals on plant growth [15] are also well documented. This characteristic can be interpreted as plant-derived smoke is not only responsible for promoting plant growth but also strategically involved in protecting plants against abiotic stresses.

### 3.2. Other Components

Another major germination stimulant was isolated from plant-derived smoke known as cyanohydrin (Table 1). Several related cyanohydrins such as mandelonitrile, acetone cyanohydrin, glycolonitrile, and 2,3,4-trihydroxybutyronitrile are present and stimulated seed germination of different plant species. The activities of these compounds are due to the spontaneous release of cyanide, suggesting an ecological role for cyanide in the post-fire revival of plant communities [21]. It was suggested that cyanide signaling interacts with the reactive oxygen species, and their effect might be linked with an obvious increase in hydrogen peroxide and superoxide anion generation [41]. Cyanide-stimulated germination is common in a wide variety of plant species [21]. The novelty of discovering cyanohydrins in smoke establishes the cyanide as an important germination stimulant in post-fire environments [21]. This clue is further strengthened by the fact that chemical compounds must be added and remain in the soil for an appropriate period for their effectiveness in post-fire environment. Cyanohydrins have a suitable time of persistence in the top layers of the soil where most of the dormant seeds are present, so they are timely available as a growth promoting signal [21]. Taking together these details, it is comprehended that further investigations are yet to be performed for more output in the form of new compounds in plant-derived smoke for its more valid implementation as growth regulator.

On the other hand, some of the compounds which are present in higher concentrations in smoke solution have inhibitory effects on plant growth [42], such as 3,4,5-trimethyl-2(5H)-furanone found in smoke solution, which inhibited seed germination [20]. This germination inhibitor is commonly present in plant-derived smoke and significantly reduced the germination-promoting nature of KAR1 [20]. The inhibitory effects of 3,4,5-trimethyl-2(5H)-furanone are only reported in lettuce seeds [20]. Furthermore, inhibitory compounds, 5,5-dimethylfuran-2(5H)-one and (5RS)-5-ethylfuran-2(5H)-one, are isolated from skilpadbessie and red oat grass-derived smoke solution, respectively [24]. To date, various compounds have been identified in plant-derived smoke (Table 1). The presence of various compounds reflects that seed cueing by plant-derived smoke is controlled by various agents; each affects the germination and growth response differently. This diverse chemistry of plant-derived smoke might be the reason behind its broad spectral impact on various growth responses of plants.

### 3.3. Plant-Derived Smoke as Biostimulant

Bio means life and stimulant means to enhance. Based on the available scientific literature and materials dedicated to this term, the biostimulants are defined as the substances (derived from living organisms by their death/decay or burning) or microorganisms that are applied to seeds, soil, plants, or on rhizosphere resulting in the stimulation of natural processes to enhance the plant quality and yield through nutrient uptake, nutrient use efficiency, and tolerance to abiotic stress [43]. In other words, biostimulants are all those products that reduce the need for synthetic fertilizers and increase plant growth and tolerance to abiotic stresses. These substances are efficient in small concentrations, favoring an efficient performance of bioactivities thus leading to high yields and good quality products. These substances offer a potentially new strategy for the regulation of physiological processes in plants leading to enhanced growth and mitigation of abiotic stresses. Keeping in view the definition, effect, and association with enhanced plant growth and stress tolerance, plant-derived smoke has recently been discussed as a “Biostimulant” [44]. Along with other molecules with biostimulant characteristics, smoke water and karrkins were used to promote the cultivation of *Amaranthus hybridus* and resulted in an increased length and fresh weight of root and shoot; increased number of leaves, total leaf area, and stem thickness as compared to non-treated plants. Protein, carbohydrate, and chlorophyll contents were also increased in response to these biostimulants, clearly emphasizing their use in agriculture. In the continuation of this research, [45] reported that among all the tested biostimulants, smoke water showed maximum enhancement of seed germination percentage in the dark. Furthermore, they suggested plant-derived smoke as the best organic biostimulant for the potential application to improve seed germination and growth of *Cleome gynandra.*

The role of plant-derived smoke in inducing abiotic-stress tolerance makes it further fit into the category of biostimulants. Among the various roles of plant-derived smoke in plant growth [25,31,32,33,34,37,38], the induction of stress tolerance that includes alleviation of salt stress in rice and maize [13,39], heat stress in tomato [38], flooding stress in soybean [14], and reversion of abscisic acid stress in lettuce [46] has highly validated its candidature as a biostimulant.

## 4. Effect of Plant-Derived Smoke on Dormancy Releasing, Seed Germination, and Seedling Growth

The majority of the research work performed on plant-derived smoke is conducted on morphological aspects of seed germination and plant growth. These investigations are about the smoke-solution effects on seed dormancy, seed germination, seedling length, and biomass. Various reports clarified that although plant responses towards smoke vary greatly, they share few common responses. This variety of responses could also be interpreted in terms of the diverse nature of chemical compounds present in smoke and their mode of application on plants. Smoke or the derived active compounds from smoke significantly enhanced germination, seedling vigor, and seedling mass of large number of plant species (Table 2, Appendix A).

### 4.1. Seed Germination

Smoke is a germination cue, and its positive effects are reported on different plant species belonging to genera and families of gymnosperms and angiosperms [98], commercial crops, and different medicinal plants [99]. Seed germination is a phenomenon in which an embryo present within the seed grows into a seedling and subsequently a plant. Various approaches have been used demonstrating the promoting effects of plant-derived smoke solution in enhancing seed germination, breaking seed dormancy, and subsequent growth processes. Explaining the possible reason involving this germination promoting and breaking seed dormancy phenomenon, it was reported that plant-derived smoke tends to induce the activity of amylase, and increase DNA replication and β-tubulin accumulation before the radicle protrusion of dormant oat caryopses, thus leading to the aforementioned processes [68]. Following similar mechanism to increase the seed germination, the positive effect of plant-derived smoke on various crops such as maize [3,12,100], rice [13,101], and wheat [102] has been clarified.

Plant-derived smoke broke seed dormancy of California chaparral plants [10], 7 species from South Western Australia [103], *Apium graveolens* [104], and wild oat [105]. Scarifications by using smoke enhanced the number and size of spreading channels in the cuticle of seeds [10]. Using a transcriptomic technique, a study on molecular aspects of seed germination in lettuce reported that abscisic acid, seed maturation, and dormancy-related transcripts were up-regulated by trimethyl butenolide and suppressed by KAR1 [106]. This study clarified that increased seed germination by KAR1 might be due to suppression of abscisic acid and dormancy-related transcripts by KAR1 present in smoke. Another investigation explaining the physiology of breaking seed dormancy in response to smoke water and KAR1 treatment was carried out by Gupta et al. [81]. It was demonstrated that smoke water and KAR1 significantly promoted the lettuce seed germination by reducing abscisic acid level and increasing the activity of hydrolytic enzymes, which supports the mobilization of stored food reserves. Although these observations are in accordance with that of Plazek et al. [72], the response mechanism of arabidopsis towards karrikins presented a somewhat different picture with independence of the abscisic acid level and requiring gibberellic acid synthesis and light [107]. The findings from these studies indicated that the response mechanism of plants towards smoke and smoke-derived compounds may vary from plant to plant. It is also reflected that plant-derived smoke not only affects the absolute concentrations of growth relating hormones but affects the fate of various physiological processes by interfering with the comparative ratio among these growth regulators that might lead to breaking seed dormancy and promoting seed germination.

### 4.2. Post-Germination Responses of Plant Towards Plant-Derived Smoke

Plant-derived smoke solution has post-germination effects. Smoke solution has positive effects on seedling development, seedling vigor, seedling length, seedling mass, and crop yield of different plants—including celery [104], okra [108], *Acacia* [109]; seedling emergence of the mesic grassland in South Africa [79]; tomato [110]; germination of Lamiaceae Mediterranean plants, slime lilies, and blouberglelie [55]. It has positive effect on root initiation of mung bean [111], tomato [110], and pollen germination and pollen tube growth of different plants [89], seedling length of Brazilian savannas [112], seedling fresh weight of perennial forage species [83], and somatic embryogenesis of tassel rope-rush [113].

Positive effects of plant-derived smoke solution at post-germination growth were observed in rice [13,114,115], wheat [102], and maize [3,12]. These reports clarified that stimulatory effects of smoke continue in post-germination stages and may enhance the biomass of the plants. The post-germination studies of smoke-treated seeds elucidate that although smoke treatment may not affect the germination stage, it still may promote growth at the post-germination stage [75]. It is, therefore, recommended to extend all germination studies necessarily up to the seedling stage so that the post-germination effects of plant-derived smoke can be interpreted.

## 5. Physiological Responses of Plant to a Plant-Derived Smoke

Plant responses to plant-derived smoke at the physiological level have been assessed for various plants. Although physiological and biochemical attributes of plants may vary from plant to plant and based on the mode of application of smoke to plants; however, they also share various responses in common. Here, various modes of smoke application on different plants and plant responses are described in detail. Several plants including maize, wheat, rice, and tomato exhibited an increased growth and vigor in response to plant-derived smoke (Table 3).

### 5.1. Pigments

Enhancement of plant growth and development in response to smoke solution are dependent on better biochemical activities, which encourage vegetative and reproductive growth. Pigments play a key role in the photosynthesis process through which plants synthesize their organic food. Pigments capture sunlight energy and convert it to chemical energy through photosynthesis. Plant-derived smoke solution-treated seedlings are vigorous and healthy [121] resulting in more chlorophyll and improved photosynthetic activity. A significant increase in the relative abundance of various photosynthetic pigments was observed in smoke-treated wheat seedlings [33]. Synthesis/accumulation of chlorophyll a relative to chlorophyll *b* increased in all the smoke treatments, resulting in increased efficiency of light reaction in plants. Plant-derived smoke solution increased biosynthesis of ribulose-1,5-bisphosphate carboxylase/oxygenase in chickpea and maize [73,74]. Smoke solution enhanced gaseous exchange, photochemical activities, and CO_2_ fixation which induced an increase in photosynthesis. An enhancement of maximum quantum efficiency, stomatal conductance, intercellular CO_2_ Concentration, and net photosynthesis was recorded in carrot [122]. These observations further prove that plant-derived smoke and smoke-derived compounds hold promising use in crop improvement in terms of photosynthetic products.

Smoke solution increased photosynthetic rate, transpiration rate, and stomatal conductance of dyer’s woad seedlings, indicating that the treatments of smoke solution could enhance the stomatal opening [48]. Plant-derived smoke solution plays a role in stomatal opening, which may be a cause of increasing photosynthetic activity. Stomatal opening is mainly controlled by abscisic acid [123]. They measured the phytohormones concentrations in germinating soybean seedlings and reported that karrikins were responsible for altering the gibberellic acid/abscisic acid ratio. These reports support that the interaction of phytohormones with smoke water and KAR1 could be anticipated as a possible mode of action for stomatal opening and other smoke/karrikin-related growth responses [123]. In rice, smoke solution enhanced the amount of chlorophyll *a*/*b* and total carotenoids with a key role in photosynthesis [13]. This increase in plant pigments enables them to harness more sunlight, thus resulting in the increased photosynthetic output of the plant. An increased response in photosynthetic pigment concentration has strengthened the fact that plant-derived smoke is equivalently active as other plant growth hormones, which take this route for plant growth promotion.

### 5.2. Phenolic Compounds

Phenolic compounds are important secondary metabolites playing a vital role in plant survival through protection against ultra-violet radiation [124], as signaling molecules [125,126] and defense against plant herbivores and insects [127]. A higher concentration of flavonoids and total phenolic contents in plants shows more antioxidant activities [128]. Smoke solution and potassium application increased secondary metabolites (flavonoids and total phenolic contents) in medicinal plants [54]. Smoke solution up-regulated phenyl propanoid pathway and flavonoid-related genes, thereby enhancing phenolic biosynthesis [73,129]. Phenolics, flavonoids, and condensed tannins were significantly increased by plant-derived smoke solution in Wild garlic [54]. The level of phytochemicals in plants is often associated with resultant biological activities. The stimulatory effects of smoke solution on photo chemicals such as indigo, phenolics, and flavonoids in several other plant species have also been reported [48,87,88]. Besides phenolics contents, plant-derived smoke solution promoted the biosynthesis of protein by increasing absorption of growth nutrients [13].

Smoke significantly increased levels of nitrogen contents in roots and shoots of papaya [51]. It was observed that application of nitrates enhanced seed germination in the fire annuals whispering bells and scorpion weed [130]. It was further assumed that oxidizing gases in smoke and acids generated on burnt sites might play a role in the germination of post-fire annuals in chaparral [131]. Nitrate was suggested to be the principal factor inducing germination by acting as a nutrient and signal well [132]. It was inferred from the existing facts that in arabidopsis, a transcription factor acts downstream of nitrate signaling which induces nitrate-dependent gene expression. This regulation triggers a nitrate-induced decrease in abscisic acid that permits seed germination [132]. Plant-derived smoke affected various biochemical phenomena of plants resulting in the regulation of various growth parameters. Certainly, the regulation of all these growth phenomena is not as straight forward as it seems, and researchers investigating the role of smoke behind these mysteries should take heed of this.

## 6. Molecular Mechanism in Plant of Plant-Derived Smoke

Exploring the molecular mechanism of plant-derived smoke solution involved several attempts to unravel the mystery behind its mechanism of action on various plant-growth processes. Plant-growth processes are controlled by various signaling molecules, which may act singly or in coordination with other molecules endogenously and exogenously, and resulted in a variety of responses. As previously discussed, the mystery as to whether plant-derived smoke is of a diverse nature and composed of a variety of compounds makes the response mechanism of plants towards smoke more complicated and diverse. Molecular studies of plants treated with smoke solution started recently and are helpful in understanding the mode of smoke solution action (Table 4).

### 6.1. Karrikins Signaling and Changes at Transcription Level

A differential expression of several expressed sequence tags related to germination, cell wall expansion, translational regulation, cell-division cycle, metabolism of carbohydrates, and regulation of abscisic acid was observed in smoke-treated lettuce [133]. Further validation of results using Real-time polymerase chain reaction presented that smoke treatment upregulated the transcript abundance of the genes, short-chain dehydrogenase/reductase, and late embryogenesis abundant protein as compared to control achenes kept in the light. Extending these investigations to maize, it was observed that smoke induced abscisic acid and other stress-related responses at the post-germination phase [134], which may result in a better adaptation to abiotic stresses. A microarray analysis was carried out in *Arabidopsis* seeds imbibed with KAR1, which resulted in an increased expression of light-induced genes revealing that karrikins have a role in priming light responses in the growing seedling [95]. Smoke solution increased the protein ubiquitination; however, KAR1-treated kernels showed different response where an aquaporin gene was up-regulated [135]. The diversity in plant responses towards smoke and karrikins is interpreted as karrikins only make up a part of the whole chemistry of smoke, which contains karrikins along with thousands of other compounds whose effects are yet to be investigated.

The mechanism of karrikin action was established in *Arabidopsis* [141]. It was demonstrated that KAR1 is perceived by its receptors, the α/β-hydrolase KAI2 (Karrikin Insensitive 2), resulting in a conformational change in KAI2. Consequently, the activated KAI2 enhances interaction with MAX2 (MORE AUXILLARY GROWTH2), which is an F-box protein. This results in the establishment of a Skp-Cullin-F-box (SCF) ubiquitin ligase complex comprising SMAX1 (SUPPRESSOR OF MAX2 1). SMAX1is considered as a putative substrate in the KAI2-SCFMAX2 complex, which is degraded in 26S proteasome after polyubiquitination. Using crystallographic techniques and ligand-binding experiments for *KAI2* recognition of karrikins, it was revealed that parts of the *KAI2*–KAR1 complex seem to adjust *KAI2* interactions to the downstream component of the karrikin-signaling pathway [141]. The mutual usage of site-directed mutagenesis, protein X-ray crystallography, and active binding measurements is evident, which makes *KAI2* fit inside the signal transduction pathway of smoke/KAR1. The studies about the seeds of *KAI2* mutants were unaffected by KAR1 [96,142] representing that *KAI2* is essential for KAR1 perception. However, conformational alteration occurs in *KAI2* at the entrance of the active position due to the KAR1 binding [141]. A crevice of hydrophobic residues linking the polar edge of KAR1 and the helical domain insert suggests that *KAI2*–KAR1 generates an adjacent interface for binding signaling in a ligand-reliant way.

### 6.2. Karrikins Signaling Pathway

Strigolactones are plant hormones that hinder shoot branching. D14 (DWARF 14) hinders rice tillering and is a candidate in the branch-inhibition pathway, whereas the close homologue D14L (DWARF 14-LIKE) contributes in the signaling pathway of KARs [143]. Clearer marks are provided for the direct binding of SL to D14 and KAR to D14L using GR24 (a strigolactone analogue) and KAR1, respectively. The specific binding of these compounds was supported by the crystal structures of D14 and AtD14L, which show the central deep hydrophobic pockets capable of accommodating each ligand.

MAX2 is a central regulator of both strigolactone and KAR signaling pathways, which control many aspects of plant growth and development. KARs binding may produce conformational variations in α/β-fold hydrolase KAI2 and its link with MAX2, an F-box constituent of E3 ubiquitin-protein ligase, and in this manner, targeting the signaling repressor(s) for degradation [144]. MAX2 in turn affects the SMAX1- and SMAX1-like proteins downstream and could be the hypothetical repressors. Strigolactone and karrikins commonly involved in seed development, growth of axillary meristems, and senescence of leaves [145]. The similar F-box protein (MAX2) and α/β hydrolase fold proteins (DAD2 and KAI2) are required for the signal transduction pathways. Structure activity relationships of analogs have indicated that both strigolactones and karrikins consist of butenolide ring that is important for biological reaction, but the rest of the molecules can be considerably modified without loss of activity. The relationship examination of strigolactones and karrikins proposed their reception by an active enzyme which then reacts with the SCF complex and presumably degrades downstream target proteins [145].

Clear evidenc about the common signal transduction mechanism of strigolactone and karrikins was found in *Arabidopsis* [96,146]. In *Arabidopsis* for karrikins responses, two genes, MAX2 and KAI2, are essential. KAI2 encodes α/β-hydrolase fold protein with an analogous structure to AtD14; therefore, it is also known as AtD14-like. The binding assay and phenotype study proposed t KAI2 as putative receptor of karrikins; thus, AtD14 and KAI2 especially facilitate plant growth to strigolactones and karrikins, respectively. Analogous phenomena were involved in strigolactone signaling, a protein, termed as SMAX1. Studies reveal that this protein is responsible for suppressing karrikin signaling in *Arabidopsis* [147]. It is suggested that SMAX1, as a specific repressor of karrikin signaling, contrary to D53, was reported to play a role in seed germination and hypocotyl elongation through KAI2-MAX2 signaling, but not in shoot branching that is controlled by AtD14-MAX2 signaling activated by strigolactones. Extending the previous investigations, it was further explained that SMXL2 controls hypocotyl growth and expression of the KAR/strigolactone transcriptional markers SMAX1 along with KUF1, IAA1, and DLK2 redundantly [148]. Hypocotyl growth in the double mutant *smax1 smxl2* is insensitive to KAR and strigolactone, resulting in reduced hypocotyl elongation. These results support the model that karrikin and strigolactone responses are mediated by different subclades of the SMXL family, and further support the case for similar butenolide signaling pathways that evolved through ancient KAI2 and SMXL duplications [148]. Summarizing all evidence, it was pointed out that despite massive explanations, there is still a huge need for further investigations of this classic family [149].

In order to explain how KAI 2 operates, various *kai2* mutations were evaluated. Among those, most were unable to accumulate the protein, which, therefore, was supposed to be unstable compared to wild type. Only one mutant *(kai2-10)* formed a protein with similar stability to the wild type, although a non-functional one. The abundance of KAI2 protein in *kai2-10* compared with the wild type was different across various experiments, which was not explained further [150]. Their results demonstrated that the enzymatic and signaling functions of KAI2 can be decoupled and provide important insights into the mechanistic events that support butenoloides signaling in plants.

Two other structurally-related compounds, (5RS)-3,4,5-trimethylfuran-2(5H)-one and 5,5-dimethylfuran-2(5H)-one, were also isolated from plant-derived smoke with an antagonistic effect against KAR1 and an inhibitory effect on seed germination of light-sensitive lettuce [24]. This investigation pointed out the dual regulatory mechanism of plant-derived smoke on post-fire seedling emergence. This leads to smoke being interpreted as a wise regulator that prevents germination when the moisture is insufficient to support the newly emerged seedling growth. However, after rainfall, this restraint is overcome by diluting the inhibitory compound in smoke which is leached out while promoting compounds are retained. Although, this supposition is quite convincing, great technology-equipped research is required to analyze and validate it widely.

### 6.3. Effect on Proteins in Plants of Plant-Derived Smoke

A proteomic approach was used to identify early karrikin-response proteins, and most of karrikin-response specific proteins were photosynthesis-, carbohydrate metabolism-, redox homeostasis-, transcription control-, transport-, and protein degradation-related proteins in *Arabidopsis* seedlings [136]. However, because there are many questions on whether karrikins are main compounds, the usage of plant-derived smoke solution is important (Figure 3).

Using the gel-free/label-free proteomic technique, proteins related to signaling and transport increased; however, protein metabolism-, cell-, and cell wall-related proteins decreased in chickpea [73]. The sucrose synthase for starch degradation and proteins for the nitrate pathway increased resulting in an increase in total soluble sugars and nitrate contents, respectively. This increase in plant growth-related proteins is a reinforcement of the fact that plant-derived smoke improves the early stage of growth in chickpea with the balance of many cascades such as glycolysis, redox homeostasis, and secondary metabolism. Additionally, proteins related to sucrose synthase, nucleotides, signaling, and glutathione significantly increased; however, cell wall-, lipid-, photosynthetic-, and amino acid degradation-related proteins decreased in maize [74]. These results suggested that increases in sucrose synthase-, nucleotide-, signaling-, and glutathione-related proteins combined with regulation of reactive oxygen species and their scavenging system in response to plant-derived smoke may improve maize growth.

A proteomic analysis was applied on soybean seedlings recovering from flooding stress followed by treatment with plant-derived smoke solution. It was investigated that sucrose/starch metabolism- and glycolysis-related proteins were suppressed in smoke-treated flooded soybean compared to flooded soybean [14]. A supportive effect of plant-derived smoke solution on soybean growth was thus established during recovery from flooding stress by balancing sucrose/starch metabolism- and glycolysis-related proteins. Furthermore, plant-derived smoke solution might enhance plant growth through the ornithine-synthesis and ubiquitin-proteasome pathway in soybean seedlings [138]. This result indicated that plant-derived smoke induced a sacrifice-for-survival mechanism in soybean seedlings during recovery after flooding causing the inhibition of the ubiquitin-proteasome pathway leading to degradation of the root tip and promotion of the lateral root development. A positive trend of smoke-treated soybean seedlings was identified in stress tolerance against flooding. This tolerance involved an abundance of ATPase, ATP content, and activation of ascorbate/glutathione cycle in response to plant-derived smoke under flooding stress [139]. These reports concluded that plant-derived smoke promoted soybean normal growth by the regulation of nitrogen-carbon transformation through the ornithine synthesis pathway and ubiquitin-proteasome pathway (Table 4).

The impact of KAR1 on tanshinone-I production and involved signal molecules was reported in red sage [140]. It was clarified that KAR1 induced generation of nitric oxide, jasmonic acid, and tanshinone-I in hairy roots of red sage. The KAR1-induced increase in tanshinone-I was suppressed by nitrogen oxide-specific scavenger (cPTIO) and nitrogen oxide inhibitors (PBITU); jasmonic acid synthesis inhibitors (SHAM) and (PrGall). These results provided the evidence that nitrogen oxide and jasmonic acid have essential roles in KAR1-induced tanshinone-I aside from other assumptions, and it can be inferred that nitrogen oxide mediated the KAR1-induced tanshinone-I production through a jasmonic acid-dependent signaling pathway which ultimately resulted in hairy roots of red sage. Modifications in gene expression, proteomic responses, and changes in tanshinone-I production and involved signal molecules in smoke-treated plants under both favorable and unfavorable conditions enrich the evidence leading to the use of plant-derived smoke as a biostimulant. A significant tolerance to abiotic stresses was reported [151] explaining that karrikin-KAI2 signaling provides arabidopsis seeds with tolerance to abiotic stress and inhibits germination under conditions unfavorable to seedling establishment. This wise regulation activity induced by plant-derived smoke changes smoke/karrikins from being a positive regulator of germination to an inhibitor resulting from the transcription of genes encoding stress response transcription factors such as *WRKY33*, *ERF5*, and *DREB2A* in a KAI2-dependent manner. However, the mechanistic details behind this mysterious reversal are yet to be understood [151]. Taken together, the results infer that smoke treatment enables plants to adopt various strategies under unfavorable conditions for protection against abiotic stresses. These strategies may involve suppression/halting the growth or promoting the growth-related activities to successfully endure the stress period [139,151,152]. This is evident from the aforementioned facts that exogenous application of plant-derived smoke ameliorates flooding-induced damages in soybean. This potential of plant-derived smoke treatment for extenuation of the flooding stress effect on plants shows its worth for stress amelioration.

## 7. Conclusions

Plant-derived smoke solution has established an incredible role in breaking seed dormancy, accelerating seed germination, and increasing seedling vigor. Previous studies resulted in the identification of numerous novel karrikins, smoke-responsive genes, and proteins that provide novel targets for detailed mechanistic studies using mutants and transgenic plants. Plant-derived smoke has conferred tolerance to plants against abiotic stresses that is attributed to its role in the regulation of the antioxidant system of plants. The role of plant-derived smoke in growth regulation under normal conditions and abiotic stresses has authenticated its candidature as the most economic and convenient biostimulant for farmers. Despite improvement in understanding the response mechanisms of plants regarding plant-derived smoke, more in-depth interdisciplinary research with a combination of omics, inter-system combinations of various scientific expertise, and interconnecting biological approaches is needed to resolve the whole chemistry and perception/signaling pathways of karrikins/plant-derived smoke. Especially interrelating omics technologies are to be performed to trace the independent response mechanism of plants irrespective of the presence of other growth regulators. A further task will be to clarify the extent of KAI2-related phenomena and how they facilitate various growth processes. The fact that whether karrikins themselves are metabolized or just triggering a series of reactions resulting in growth regulation is yet to be unraveled. Furthermore, independent separation of promoting and inhibitory compounds is required for the explanation of their specific correlation with growth phenomena that we aimed for. Based on this information, plant-derived smoke applications should also be performed in the laboratory and field before accepting them at a wide agricultural level to optimize their benefits.

## Figures and Tables

**Figure 1 ijms-21-07760-f001:**
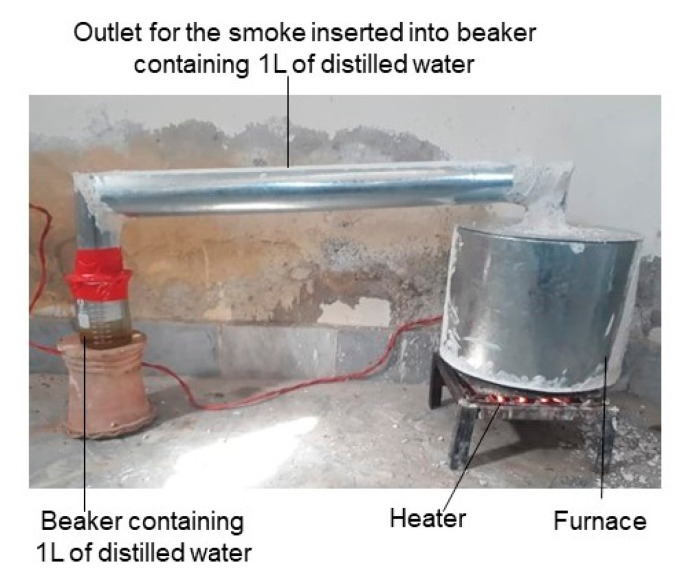
Preparation of plant-derived smoke solution. Plant material is burnt in the specified chamber through an electric heater. The smoke coming from burning plant material is collected through a pipe by bubbling this smoke into 1 L distilled water at the other end of pipe. All the setup is completely sealed to avoid any smoke escape.

**Figure 2 ijms-21-07760-f002:**
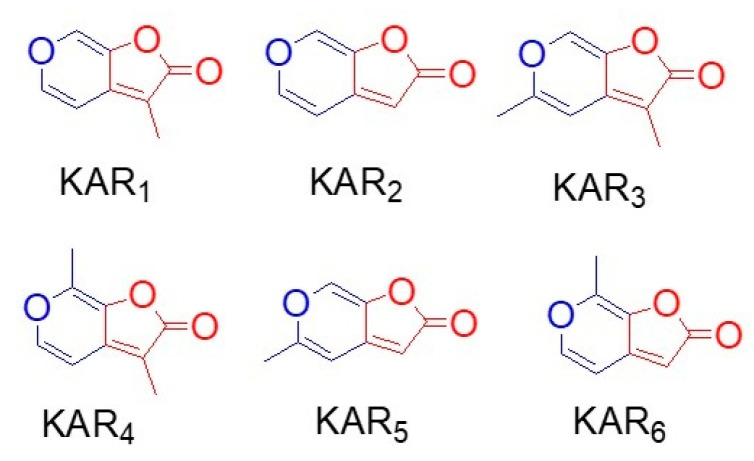
Chemical structures of karrikins family involved in seed germination. KAR1, KAR2, KAR3, KAR4, KAR5, and KAR6 were shown. All karrikins share the basic structure commonly with two ring structures: one is a six-membered heterocyclic pyran shown as blue and the other is a five-membered lactone ring as a butanolide shown as red.

**Figure 3 ijms-21-07760-f003:**
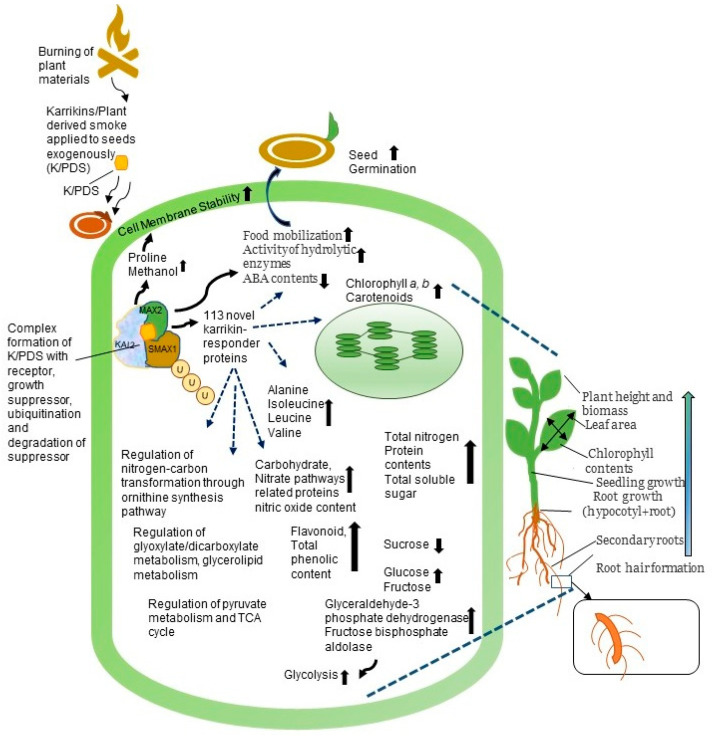
Schematic diagram showing diverse cellular responses to plant-derived smoke. Plant-derived smoke triggers regulation of various metabolic processes resulting in acceleration of seed germination and other growth processes. Upward and downward arrows indicate increased and decreased metabolic processes in response to plant-derived smoke, respectively. The dashed lines represent the possible relationship between smoke induced changes at molecular level and their effects. Abbreviations are as follows: K/PDS, karrikins/plant-derived smoke; KAI2, α/β hydrolases receptors of karrikins; MAX2, F-box subunit of a SCF class of E3 ubiquitin-protein ligase complex; SMAX1, a growth suppressor protein; and TCA, carboxylic acid cycle.

**Table 1 ijms-21-07760-t001:** A list of different compounds identified in plant-derived smoke.

Plant Source for Plant-Derived Smoke	Identified Compounds	References
Lettuce	1,8–cineole	[19]
Skilpadbessie; Red oat grass	3-methyle-2-Hfuro [2,3-C]pyran-2-one (Butenolide)	[11]
Skilpadbessie; Red oat grass	3,4,5-trimethylfuran-2(5H)-one	[20]
Red-and-green kangaroo paw	glyceronitrile, cyanohydrin	[21]
Coyote tobacco	Catechol	[22]
Ginkgo	Hydroquinone	[23]
Skilpadbessie	5,5-dimethylfuran-2(5H)-one	[24]
Red oat grass	(5RS)-5-ethylfuran-2(5H)-one	[24]

**Table 2 ijms-21-07760-t002:** Effect of plant-derived smoke on seed germination and plant-growth responses of various plant (2010–2020).

Experimental Plant Species	Major Findings	Ref.
**Smoke solution for application**		
Lantana	Seed germination, germination velocity index and vigor index increased	[47]
Dyer’s woad	Seedling mass increased	[48]
Rock rose	Germination percentage increased	[49]
Edible banana	Seedling length, seedling mass, number of shoots, number of roots, number of leaves, leaf area increased	[36]
Plant species from grassland	Seed germination increased	[50]
Pawpaw	Seed germination rate, seedling length and vigor, and number of leaves increased	[51]
Canola	Plant regeneration, seedling length increased	[52]
Mediterranean Basin flora	Seedling emergence was promoted	[53]
Society garlic., Wild garlic	Seed germination, seedling mass, root length and root number increased	[54]
*Lavandula stoechas,* *Origanum onites, Phlomis bourgaei, Stachys cretica, Satureja thymbra, Teucrium lamiifolium*	Seed germination increased	[55]
Coyote tobacco., Whispering bells	Germination percentage increased	[56]
Sweet potato	Number of adventitious roots, length of adventitious roots and length of lateral roots increased	[57]
Millet	Seed germination, seedling length and seedling mass were enhanced	[58]
Carrot	Seed germination, seedling length increased	[59]
Barnyard grass	Germination percentage, relative root elongation, seedling length and seedling mass were promoted	[60]
Perennial forage species	Seed germination increased	[61]
12 eastern Mediterranean basin plants	Seed germination increased	[62]
10 Interior West Penstemon species	Seed germination increased	[63]
Canadian horseweed	Seed germination, seedling growth increased	[64]
Wheat	Germination percentage, germination index, seedling vigor index and seedling length increased	[33]
*Helianthemum tinetense*	Seed germination increased	[65]
*Astragalus verus, Bromus tectorum*	Seed germination increased	[66]
Mediterranean Basin flora	Seed germination increased	[67]
Wild oat	Germination percentage and per unit weight water content increased, coat rupturing was stimulated	[68]
Shortgrass Prairie species	Seed germination increased	[69]
Mediterranean plant species	Seed germination and seedling length were enhanced	[70]
Rice	Root length and root fresh/dry weights increased	[15]
Cape flats sand Fynbos species	Seedling length and seed germination were promoted	[71]
Wheat	Root/shoot length, root fresh/dry weight, shoot fresh/dry weight and leaf area increased	[33]
*Lupinus angustifolius*	Seed germination increased	[72]
Chickpea	Seed germination, seedling length and mass increased	[73]
Maize	Seed germination, seedling length and mass increased	[74]
Tomato, Cucumber, Pot marigold, Sword lily	Seed germination percentage/rate, seedling length and fresh weight increased	[35]
Lettuce	Seed germination percentage was promoted	[75]
*Calotropis gigantea*	Seed germination increased	[76]
Rice	Seed water uptake and germination percentage were enhanced	[77]
**Trimethylebutenolide analogs and butenolide solution**		
Lettuce., Whispering bells., Tomato bush	Germination percentage increased	[29]
Lettuce	Germination percentage increased	[78]
Lettuce	Germination percentage increased	[79]
Tangle head	Germination percentage increased	[80]
Lettuce	Seed germination percentage was promoted	[81]
**Smoke and butenolide solution**		
onion	Number of leaves, leaf length, leaf weight, bulb diameter and bulb weight increased	[35]
Melon	Seedling mass increased	[82]
Button creeper	Seed germination increased, and seed dormancy broke	[83]
Asian mustard	Germination percentage increased	[84]
Tropical soda apple	Seed germination, seedling length and mass increased	[85]
Wild oat, Wimmera ryegrass, Weeping lovegrass, Little seed canary grass, Barley grass, Perrenial veldgrass, Ripgut brome	Germination percentage increased	[86]
Edible banana	Leaves number, branching, seedling length, seedling weight and root number increased	[87]
Tree aloe	Seed germination and seedling growth increased	[88]
Torch lily, Opal flower	Pollen germination and pollen tube growth was enhanced	[89]
Kikuyu grass	Seedling vigor, seedling mass, and leaf number increased	[90]
Okra	Seedling length increased	[91]
Lettuce	Seed germination and radicle length increased	[92]
**Glyceronitrile and smoke/butanolide solution**		
Kangaroo paw., *Gyrostemon*, *Racemigerus*, *Gyrostemon ramulosus*	Seed germination and seedling length were enhanced	[93]
Kangaroo paw	Seed germination and embryo growth increased	[94]
Mediterranean plant species	Seed germination and seedling length were enhanced	[70]
**Aerosol smoke and smoke solution**		
*Arabidopsis*	Seed germination and hypocotyl length increased; and seed dormancy was broken	[95,96]
**Smoke and PGPR solution**		
Rice	Seed germination and shoot/root lengths increased	[97]

Ref.: References.

**Table 3 ijms-21-07760-t003:** Effect of plant-derived smoke on physiological responses of various plants.

Mode of Smoke/Smoke Compounds Application	Major Findings	Ref.
**Tomato**		
Seed imbibition in Butenoloides solution	Total soluble proteins in embryo, cotyledons and seedlings increased	[116]
Smoke and butenolide solution	Ascorbic acid, b-carotene, lycopene and total soluble solids increased	[34]
Smoke solution prepared from different plants species	α-amylase activity and abscisic acid content, N, P, and K ion contents, chlorophyll contents increased	[35]
**Rice**		
Seed priming in smoke solution	Proline contents, photosynthetic pigments increased	[101]
Seed priming in smoke solution	Ion contents, cell membrane stability, protein contents, total nitrogen contents increased	[115]
**Wheat**		
Smoke solution	Electrolyte (Na^+2^, Ca^+2^, K^+^) contents, nitrogen and protein contents, total soluble sugar, total soluble proteins, proline contents, glycine betaine, antioxidant enzymes increased	[33]
Smoke and PGPR solution	Photosynthetic pigments, Electrolyte (Na^+2^, Ca^+2^, K^+^) content, semyzne tnadixoitna, enilorp, stnetnoc nietorp, ragus elbulos latotdesaercni	[102]
Smoke solution	Carbohydrate, protein and lipid analysis, macro and micro elements concentrations increased	[33]
**Wild garlic**		
Smoke solution	Flavonoids, total phenolics, condensed tannins were regulated	[54]
**Dyer’s woad**		
Smoke solution	Indigo concentration increased	[48]
Smoke solution	Photosynthetic yield, chlorophyll fluorescence increased	[117]
**Kikuyu grass**		
Smoke and butenolide solution	Cd uptake decreased	[90]
**Pawpaw**		
Smoke solution	Nitrogen, Ion contents, Fe, Zn, Cu, chlorophyll content increased	[51]
**Tree aloe**		
Smoke and butenolide solution	Flavonoids, total phenolics were regulated	[88]
**Bone seed**		
Aerosol smoke solution application	Stomatal conductance, CO_2_ assimilation rate and intercellular CO_2_ levels increased	[118]
**Edible banana**		
Smoke and karrikinolide (butenolide) solution	Photosynthetic pigments, total phenolics, total flavonoids, proanthocyanidins increased	[36]
**Ear-leaf nightshade, Talinum, Asthma-weed, Catsear, Akmella**		
Smoke, butanolide and trimethylebutanolide solution	α amylase activity increased	[119]
**Okra**		
Smoke, butanolide and trimethylebutenolide solution	α amylase activity and bacterial abundance increased	[119]
**Maize**		
Smoke solution priming	Ion contents, photosynthetic pigments and antioxidant enzymes increased	[39]
Smoke solution	Chlorophyll pigments and total soluble proteins increased	[12]
**Lettuce**		
Smoke solution	Total soluble sugar increased	[120]
Smoke/butenolide/ trimethylebutanolide solution	α- amylase activity, starch, sugar, protein contents, lipase activity and lipid contents increased	[81]
**Barnyard grass**		
Smoke solution	α amylase and abscisic acid contents increased	[60]
**Sword lily, Cucumber, Pot marigold**		
Smoke solution prepared from different plants species	α-amylase activity and abscisic acid content, N/P/K ion contents, and chlorophyll contents increased	[80]
**Wild oat**		
Smoke solution	α/β-amylase activities, starch contents, β tubulin accumulation increased	[68]
**Chickpea**		
Smoke solution	Total soluble sugar, total soluble proteins, number of rhizobia increased	[73]

Ref.: References.

**Table 4 ijms-21-07760-t004:** Summary of molecular analyses of various plants exposed to karrikins/butenolides and plant-derived smoke.

Smoke Compounds	Major Findings	Ref.
**Lettuce**		
Smoke solution	Genes related to germination, cell wall expansion, translation, cell division cycle, carbohydrate metabolism and abscisic acid regulation were regulated	[133]
KAR1, trimethylbutenolide	Abscisic acid, seed maturation and dormancy-related transcripts were up-regulated by trimethylbutenolide and suppressed by KAR1	[106]
**Maize**		
Smoke solution	Stress- and abscisic acid-related genes were up-regulated	[134]
Smoke solution, KAR1	Smoke-water enhanced the ubiquitination of proteins and activated protein-degradation-related genes. KAR1 up regulated aquaporin gene	[135]
Smoke solution	Sucrose synthase-, nucleotides-, signaling-, and glutathione-related proteins increased; cell wall-, lipid-, photosynthesis-, and amino acid degradation-related proteins decreased	[74]
***Arabidopsis***		
Karrikins	Karrikin signaling is F-box protein (MAX2) dependent. Seed germination and seedling photomorphogenesis was triggered by karrikin	[96]
Karrikins	Photosynthesis, carbohydrate metabolism, redox homeostasis, transcription control, protein transport, processing, protein degradation were regulated	[136]
***Salmonella typhimurium***		
Smoke solution, 3,7-dimethyl-2H-furo[2,3-c] pyran-2-one	No genotoxicity from smoke solution and smoke isolated compounds	[137]
**Tomato**		
Butenolides	Butenolides changed the DNA, RNA and protein profiles, no effect on integrity of DNA	[116]
**Chickpea**		
Smoke solution	Signaling-, nitrate pathway-, and transport-related proteins increased. Protein metabolism-, cell-, and cell wall-related proteins decreased	[73]
**Soybean**		
Smoke solution	Protein abundance and gene expression of O-fucosyltransferase family proteins increased, while that of peptidyl-prolyl cis-trans isomerase and Bowman-Birk proteinase isoinhibitor D-II decreased, sucrose/starch metabolism and glycolysis were suppressed	[14]
Smoke solution	Proteins related to protein synthesis, arginine metabolism and ubiquitin-proteasome pathway were regulated; metabolites related to amino acid, carboxylic acids, and sugars were mostly altered	[138]
Smoke solution	Protein metabolism-, stress-, redox-, and mitochondrial electron transport chain-related proteins were regulated	[139]
**Red sage**		
KAR1	Production of tanshinone-I increased	[140]

Ref.: References.

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
