# Peer review of "Plant-Derived Smoke Affects Biochemical Mechanism on Plant Growth and Seed Germination"

_ijms, 2020, doi:10.3390/ijms21207760_

Round 1
Reviewer 1 Report
Dear Editor:
The manuscript has some structure and adaptation errors (among others) to the journal that I am going to detail, as an example:
- Title:
- a) The title is very broad, and most of the manuscript deals with general questions (especially the first 5 sections) that could be far from the journal's objectives.
- b) The greatest effect of these substances is focused on germination. Possibly, the authors should reorient their manuscript to the study of germination.
- c) The authors should reconsider whether the effect of these substances corresponds to the effect of a biostimulant. Although, this action is not reflected in the title.
- The authors do not delve into the effects of these substances and they write many words that could be simplified (page 4, line 110-117).
- The authors do not describe the effects of these substances in horticulture (page 4, line 120), and furthermore, they only introduce a reference for it (ref. 31).
- The authors talk about different species of plants (page 4, line 123), but they do not specify any. Furthermore, they do not indicate any reference.
- The manuscript refers to the effect on abiotic stress, but it does not refer to the mechanisms responsible for this effect.
- If these substances have an effect against abiotic stress, they should be considered biostimulants. The authors can read the references on these substances.
- In general, the authors indicate the effect on germination, but they do not explain how this effect is in plants. The authors dedicate a table to highlight the effect on germination, but they do not detail the modified mechanisms in the plant that favor germination.
- The manuscript refers to the antioxidant effect of cyanide, but it does not indicate how this effect is.
- There are many claims that the author is unknown (example: page 4, line 138-140).
- The authors include the dormancy of the seeds, the germination of the seeds, the length of the primordia, and the biomass, as morphological effects in section 4. It is a very general approach because the dormancy of the seeds should be studied from a physiological point of view.
- Instead, the authors indicate (page 8, line 202) the effect on growth and vigor in section 5 (physiological response). Although, the growth of the plant has also been discussed in section 4.2. It seems confusing.
- In this section 5, only reference is made to pigments and phenolic compounds. The effect on dormancy at the physiological level should be explained.
- The authors talk about stomatal opening, but they do not detail how stomata open and its reason (page 10, line 217).
- On page 11, line 239-240, the authors state generally for all crops or plants with reference 88 (on papaya only). The same goes for the next two sentences. If the authors indicate that nitrates break dormancy they should explain this effect (page 11, line 241.
- At the molecular level, the authors dedicate only 4 pages of the 16 pages of the manuscript.
- The first conclusion (page 15, line 387-388) does not correspond to the title. Although, it is in accordance with the manuscript. However, the authors have gone into little depth on this issue.
- The conclusions do not refer to the antioxidant effect and the biostimulant effect.
- Finally, the conclusions do not refer to molecular mechanisms.
In relation to these errors, I consider the text to be confusing and messy. Authors should ponder these words and address their hypothesis in greater scientific depth.
Author Response
The manuscript has some structure and adaptation errors (among others) to the journal that I am going to detail, as an example:
Answer: Thank you very much for your correction and comments. Based on three reviewers’ comments, this article has been improved. Please review this article, again.
- Title:
Answer: Contents have been corrected based on comments from reviewer. And then, title has been changed as follows: “Plant-Derived Smoke Affects Biochemical Mechanism on Plant Growth and Seed Germination”
- a) The title is very broad, and most of the manuscript deals with general questions (especially the first 5 sections) that could be far from the journal's objectives.
Answer: Thank you very much for severe comments from reviewer. Based on the comments from reviewer, all parts have been carefully corrected and re-organized.
- b) The greatest effect of these substances is focused on germination. Possibly, the authors should reorient their manuscript to the study of germination.
Answer: All parts have been corrected carefully based on reviewers’ comments; however, one part has been left, which is the session of plant growth. Because our main target is not only seed germination but also plant growth, the explanation of plant growth has been left. Based on this reason, contents in this review article have been added the research on plant growth.
- c) The authors should reconsider whether the effect of these substances corresponds to the effect of a biostimulant. Although, this action is not reflected in the title.
Answer: Thank you very much for this nice suggestion, explanation has been added on the plant-derived smoke in terms of biostimulant and relevant literature has been added in detail.
- The authors do not delve into the effects of these substances and they write many words that could be simplified (page 4, line 110-117).
Answer: Sorry for such mistake. The text has been simplified and marked as red color in the section “3.1 Karrikins”.
- The authors do not describe the effects of these substances in horticulture (page 4, line 120), and furthermore, they only introduce a reference for it (ref. 31).
Answer: A detailed account of effect of plant-derived smoke on horticultural crops has been added and marked as red color in the section “3.1 Karrikins”.
- The authors talk about different species of plants (page 4, line 123), but they do not specify any. Furthermore, they do not indicate any reference.
Answer: Correction has been made and marked as red color in the section “3.1 Karrikins”.
- The manuscript refers to the effect on abiotic stress, but it does not refer to the mechanisms responsible for this effect.
Answer: A brief overview of the mechanism has been incorporated in the section “3.1 Karrikins”, however more in-depth details were avoided as purely mechanistic approach may take away from the focus of this review.
- If these substances have an effect against abiotic stress, they should be considered biostimulants. The authors can read the references on these substances.
Answer: Thank you very much. The relevant literature has in this regard has been read in detail and added in section “3.3. Plant-Derived Smoke as Biostimulant”.
- In general, the authors indicate the effect on germination, but they do not explain how this effect is in plants. The authors dedicate a table to highlight the effect on germination, but they do not detail the modified mechanisms in the plant that favor germination.
Answer: Thank you very much for your comment. A brief overview of the mechanism behind increased germination is descried and marked as red color in the section “4.1. Seed Germination”.
- The manuscript refers to the antioxidant effect of cyanide, but it does not indicate how this effect is.
Answer: Details about the possible way that how cyanide effect has been added and marked as red color in the section “3.2. Other Components”.
- There are many claims that the author is unknown (example: page 4, line 138-140).
Answer: Sorry for that mistake; reference has been added and marked as red color in the section “3.2. Other Components”.
- The authors include the dormancy of the seeds, the germination of the seeds, the length of the primordia, and the biomass, as morphological effects in section 4. It is a very general approach because the dormancy of the seeds should be studied from a physiological point of view.
Answer: Thank you very much. The dormancy of the seeds and germination of the seeds with physiological point of view has been explained and marked as red color in the section “4.1. Seed Germination”.
- Instead, the authors indicate (page 8, line 202) the effect on growth and vigor in section 5 (physiological response). Although, the growth of the plant has also been discussed in section 4.2. It seems confusing.
Answer: Correction has been made in title of section “4.2. Post-Germination Responses of Plant Towards Plant Derived-Smoke”. In section “4.2. Post-Germination Responses of Plant Towards Plant Derived-Smoke”, growth was discussed just to germination and post-germination stages, but in Section “5. Physiological Responses of Plant to a Plant-Derived Smoke”, the broader aspects of later growth responses of plants towards smoke are discussed.
- In this section 5, only reference is made to pigments and phenolic compounds. The effect on dormancy at the physiological level should be explained.
Answer: A detail account has been added in section “5. Physiological Responses of Plant to a Plant-Derived Smoke”.
- The authors talk about stomatal opening, but they do not detail how stomata open and its reason (page 10, line 217).
Answer: The details have been described and marked as red color section “5.1. Pigments”.
- On page 11, line 239-240, the authors state generally for all crops or plants with reference 88 (on papaya only). The same goes for the next two sentences. If the authors indicate that nitrates break dormancy, they should explain this effect (page 11, line 241.
Answer: Thank you very much for pointing this correction. Plant name is specified and role of nitrates in breaking dormancy has been explained in detail in section “5.2. Phenolic Compounds”, the added part has been marked as red color.
- At the molecular level, the authors dedicate only 4 pages of the 16 pages of the manuscript.
Answer: Following all recommendations from reviewer, plant-derived smoke as biostimulant has been discussed in section “3.3. Plant-Derived Smoke as Biostimulant ”, also molecular parts have been added staying within the spectrum of this review.
- The first conclusion (page 15, line 387-388) does not correspond to the title. Although, it is in accordance with the manuscript. However, the authors have gone into little depth on this issue.
Answer: The title has been changed.
- The conclusions do not refer to the antioxidant effect and the biostimulant effect.
Answer: The explanations of antioxidant effect and the biostimulant effect have been added in the section of “7. Conclusions”.
Finally, the conclusions do not refer to molecular mechanisms.
Answer: Molecular mechanisms have been added in the section of “7. Conclusions”.
In relation to these errors, I consider the text to be confusing and messy. Authors should ponder these words and address their hypothesis in greater scientific depth.
Answer: Based on three reviewers’ comments, this article has been improved. Please review this article, again.
Reviewer 2 Report
The study by Khatoon et al. describes a latest research explaining the effect of plant-derived smoke on morphological, physiological, biochemical, and molecular responses of plants.
The work is interesting and clear, but does not present new data - there are many reviews from the last 10 years, to which this work is only similar.
The material is very well organized in tables but poorly characterized- the text requires rebuilding and more explanation.
The title of table 2 should be changed. It indicates morphological changes, while the effect on seed germination was mainly described.
The latest data on plant responses at the molecular level were very lacking.
The number of publications cited is too long. Please refer to the most important data.
The article needs to be corrected by a native English speaker.
However, I cannot recommend the publication of this paper in the present form but it may be acceptable after revision and re-evaluation.
Author Response
The study by Khatoon et al. describes a latest research explaining the effect of plant-derived smoke on morphological, physiological, biochemical, and molecular responses of plants. The work is interesting and clear, but does not present new data - there are many reviews from the last 10 years, to which this work is only similar.
Answer: Recent work relevant to plant derived smoke has been added, i.e, Moreira & Pausas, 2018, Cox et al., 2017, Roeder et al., 2019, Catav et al., 2018, Plazek et al., 2018, Zhou et al., 2019, Akeel et al., 2019, Jamil et al., 2020, Wang et al., 2020 etc. All added parts have been marked as red color.
The material is very well organized in tables but poorly characterized- the text requires rebuilding and more explanation.
Answer: As suggested, description of tables is extended further for each table.
The title of table 2 should be changed. It indicates morphological changes, while the effect on seed germination was mainly described.
Answer: Thank you very much for your suggestion. Table 2 has been shifted as Supplemental Table 1. Using current publication (2010-2020) for seed-germination and plant-growth, Table 2 has been re-organized.
The latest data on plant responses at the molecular level were very lacking.
Answer: Respected sir, the latest data on plant responses at the molecular level has been extended and marked as red color.
The number of publications cited is too long. Please refer to the most important data.
Answer: Publications have been reduced and contents have been re-written as suggested.
The article needs to be corrected by a native English speaker.
Answer: Sorry for these mistakes. This article has been corrected by native English speaker.
However, I cannot recommend the publication of this paper in the present form but it may be acceptable after revision and re-evaluation.
Answer: Based on three reviewers’ comments, this article has been improved. Please review this article, again.
Reviewer 3 Report
The presented work for review raises an important issue related to the influence of factors on plant germination and development. In this work, the authors present the method of producing plant smoke, describe the chemical composition of the obtained plant-derived smoke and further influences on the morphology and physiology of plants. The work is interesting, but I think it is worth expanding it to include the issue of the possible presence of pollutants in the obtained fumes, which will undoubtedly affect the spread of pollutants in the environment. Additionally, I think it is worth mentioning the disadvantages of this method.
Author Response
The presented work for review raises an important issue related to the influence of factors on plant germination and development. In this work, the authors present the method of producing plant smoke, describe the chemical composition of the obtained plant-derived smoke and further influences on the morphology and physiology of plants.
The work is interesting, but I think it is worth expanding it to include the issue of the possible presence of pollutants in the obtained fumes, which will undoubtedly affect the spread of pollutants in the environment.
Answer: Thank you very much for this important point. Correction has been made in “Introduction” of the text and marked as red color.
Additionally, I think it is worth mentioning the disadvantages of this method.
Answer: Disadvantages have been described along with the answer of previous comment and marked as red color.
Round 2
Reviewer 1 Report
Now it is more complete
Author Response
There is no more suggestion from reviewer.
Reviewer 2 Report
This paper by Khatoon et al has clearly benefited from the revision, as advised by reviewers. Readability is significantly enhanced and the text is more concise.
Below are some comments that should be included in the manuscript:
Line 204. The chapter title should be changed (is different from the content in its presents form; seed dormancy releasing and seed germination are not morphological effect on plant). Suggests: 4. Effect of plant-derived smoke on dormancy releasing, seed germination and seedling growth.
Line 206 (and other). seed-germination should be changed to seed germination or germination of seeds.
Line 233-234. KAR1 induced germination of dormant Avena fatua and non-dormant Lupinus angustifolius seeds associated with the control of ABA level and increasing the activity of hydrolytic enzymes (Cembrowska-Lech et al., 2015, Journal of Plant Physiology 176:169–179; PÅ‚ażek et al., 2018, Int. J. Mol. Sci. 2018, 19(8), 2416). Interestingly, KAR1 was also able to stimulate germination of dormant Arabidopsis thaliana seeds without changing the ABA level (Nelson et al., 2009, ant physiology 149(2):863-73).
Author Response
Reviewer 2
This paper by Khatoon et al has clearly benefited from the revision, as advised by reviewers. Readability is significantly enhanced and the text is more concise.
Below are some comments that should be included in the manuscript:
Line 204. The chapter title should be changed (is different from the content in its presents form; seed dormancy releasing and seed germination are not morphological effect on plant). Suggests: 4. Effect of plant-derived smoke on dormancy releasing, seed germination and seedling growth.
Answer: Thank you very much for your suggestion. Title has ben changed with green color.
Line 206 (and other). seed-germination should be changed to seed germination or germination of seeds.
Answer: The recommended correction is done and checked carefully. The corrected part has been marked with green color.
Line 233-234. KAR1 induced germination of dormant Avena fatua and non-dormant Lupinus angustifolius seeds associated with the control of ABA level and increasing the activity of hydrolytic enzymes (Cembrowska-Lech et al., 2015, Journal of Plant Physiology 176:169–179; PÅ‚ażek et al., 2018, Int. J. Mol. Sci. 2018, 19(8), 2416). Interestingly, KAR1 was also able to stimulate germination of dormant Arabidopsis thaliana seeds without changing the ABA level (Nelson et al., 2009, ant physiology 149(2):863-73).
Answer: Thank you very much for this important point. The explanation of this varying behavior has been added in section 4.1. (Seed Germination) and marked with green color.
Reviewer 3 Report
The changes introduced by the authors significantly increased the value of the manuscript. I have no comments on this version of the work.
Author Response

(The authors gave the same response as above.)
